# Enantioselective desymmetrization of cyclohexadienones via an intramolecular Rauhut–Currier reaction of allenoates

Weijun Yao[1], Xiaowei Dou[1], Shan Wen[1], Ji'en Wu[1], Jagadese J. Vittal[1] & Yixin Lu[1]

The Rauhut–Currier (RC) reaction represents an efficient method for the construction of carbon–carbon bond in organic synthesis. However, the RC reactions involving allenoate substrates are very rare, and in particular, asymmetric intramolecular RC reaction of allenoates is yet to be discovered. Here, we show that the intramolecular RC reaction proceeds smoothly in the presence of 1 mol% β-ICD, and bicyclic lactones are obtained in high yields and with excellent enantiomeric excesses. With the employment of γ-substituted allenoates as racemic precursors, a novel dynamic kinetic resolution of allenes via RC reaction is observed, which allows for facile synthesis of highly enantiomerically enriched allenes.

[1] Department of Chemistry, National University of Singapore, 3 Science Drive 3, Singapore 117543, Singapore. Correspondence and requests for materials should be addressed to Y. L. (e-mail: chmlyx@nus.edu.sg).

ewis base-catalysed transformations play significant roles in carbon–carbon and carbon–heteroatom bond-forming reactions[1]. Allenoates are popular substrates in Lewis base catalysis, due to their ready availability and unique reactivity[2]. As illustrated in Fig. 1, addition of a Lewis base to the β-carbon of an allenoate creates a zwitterionic intermediate, which has several resonance structures, and versatility of such a zwitterionic intermediate dictates its versatile reactivity. In a common reaction pathway, the zwitterionic species is trapped by an activated alkene, and dipolar [3 + 2] cyclization then takes place to yield a five-membered ring structure (Fig. 1, pathway a). In this context, tertiary phosphines have been firmly established as effective catalysts for a diverse array of annulation reactions of allenoates in the past decades[2–4]. However, it should be noted that the Lewis base-triggered nucleophilic zwitterion may react differently (Fig. 1, pathway b): addition of activated nucleophilic species to activated olefin without subsequent cyclization would lead to an allenoate Rauhut–Currier (RC) process. The RC reaction, also known as the vinylogous Morita–Baylis–Hillman reaction, commonly refers to the coupling reaction between two active alkene reaction partners. In the presence of nucleophilic catalyst, a new carbon–carbon bond is created between the α-position of one activated alkene and the β-position of a second alkene[5–7]. The past decade has seen remarkable advance in enantioselective RC reactions, and many excellent examples describing enantioselective intermolecular[8,9] and intramolecular RC reaction[10–13] have been reported. It is noteworthy that in contrast to rich literature reports on cycloaddition reactions, there were only a few isolated examples up to date on the RC reaction involving allenoate substrates. In 1999, Lu et al.[14] investigated phosphine-catalysed [3 + 2] annulation reaction of allenoates with activated alkenes, and they observed the formation of RC by-products, in addition to the anticipated carbocyclics. In 2003, Miller et al. reported an amine-catalysed coupling reaction between allenic esters and α,β-unsaturated carbonyl compounds[15]. Recently, Shi et al.[16] disclosed an enantioselective intermolecular RC reaction of electron-deficient allenes with maleimides. To the best of our knowledge, an intramolecular RC reaction of allenes remains unknown, let alone an enantioselective variant. We were, therefore, very interested in developing such a process.

Catalytic enantioselective desymmetrization of cyclo-hexadienones[17,18], which can efficiently transform prochiral molecules into complex bicyclic architectures in one-step operation, has attracted much attention in the past decade. In addition to a number of transition metal-based approaches[19–24], enantioselective organocatalytic desymmetrizations of dienone

systems have also been developed, including: the Stetter reaction[25–29], Michael addition[30–38], the RC reaction[13–15] and cascade reaction[39].

Our group has been extensively investigating enantioselective catalytic reactions utilizing allene substrates in the past few years[40–42]. To further extend the utility of allenoates, we became interested in developing an intramolecular desymmetrization process by using allene-tethered substrates. As shown in Fig. 2, when a prochiral cyclohexadienone with an allene moiety is subjected to Lewis base activation, and the zwitterionic intermediate generated may undergo different reactions; the Lu's [3 + 2] cyclization may create a tricyclic system, and the RC-type reaction will give a bicyclic lactone with an exo-cyclic allene structure. While both pathways are feasible, we envisioned judicious selection of catalytic systems will make unprecedented RC reaction of allenoates favourable.

Herein, we document an intramolecular allenoate RC reaction, which results in enantioselective desymmetrization of cyclohexadienones and creation of biologically useful bicyclic structural motifs[43–46]. Moreover, by employing substrates with γ-substituted allene moieties, we develop a dynamic kinetic resolution (DKR) process to access chiral allenes.

## Results

**Reaction optimization.** We initiated our investigation by employing allene–dienone **1a**, and examined its potential modes of reaction (Table 1). Different phosphine catalysts were examined, and neither annulation nor RC product was observed (entries 1–6). While 4-dimethylaminopyridine was completely ineffective (entry 7), employment of 1,4-diazabicyclo[2.2.2]octane (DABCO) led to the formation of the RC product in high yield (entry 8). Subsequently, chiral amine catalyst β-ICD was chosen for the enantioselective version of the reaction. To our delight, β-ICD worked remarkably well, affording the desired RC product **2a** in good yield and high enantiomeric excess (entry 9). The catalytic effects of other cinchona alkaloids were also examined, either low reactivity or low enantioselectivity was obtained in all the cases (entries 10–15). A quick solvent screening established the best reaction conditions (entries 16–21). The catalyst loading could be reduced to 1 mol%, which made this method more appealing and practical. In the presence of 1 mol% β-ICD, the desymmetrization of cyclohexadienone **1a** took place smoothly to afford bicyclic allene **2a** in 96% yield and 96% ee (entry 22).

**Substrate scope.** Starting cyclohexadienone–allenes containing an alkyl substituent were well-tolerated; the alkyl group can be

**Figure 1 | Allenoate activations via Lewis base catalysis.** The zwitterionic intermediate generated upon the addition of a Lewis base to an allenoate can be trapped in different ways leading to the annulation (pathway a) or allenoate RC products (pathway b).

**Figure 2 | Desymmetrization of cyclohexadienones via an intramolecular process.** Intramolecular [3 + 2] annulation (**a**) and RC reaction of allenoate (**b**) are two competing pathways.

**Table 1 | Enantioselective intramolecular RC-type reaction of allenoate.**

| Entry | Catalyst | Solvent | t (h) | Yield (%)* | ee (%)† |
|---|---|---|---|---|---|
| 1 | PPh₃ | CHCl₃ | 24 | – | – |
| 2 | MePPh₂ | CHCl₃ | 24 | – | – |
| 3 | Me₂PPh | CHCl₃ | 24 | – | – |
| 4 | **3** | CHCl₃ | 24 | – | – |
| 5 | **4** | CHCl₃ | 24 | – | – |
| 6 | **5** | CHCl₃ | 24 | – | – |
| 7 | DMAP | CHCl₃ | 24 | – | – |
| 8 | DABCO | CHCl₃ | 2 | 85 | – |
| 9 | β-ICD | CHCl₃ | 1 | 84 | 92 |
| 10 | Quinine | CHCl₃ | 72 | 54 | 14 |
| 11 | Quinidine | CHCl₃ | 72 | 57 | 11 |
| 12 | Cinchonine | CHCl₃ | 72 | 55 | 50 |
| 13 | Cinchonidine | CHCl₃ | 72 | 61 | − 56 |
| 14 | (DHQ)₂PHAL | CHCl₃ | 24 | 81 | − 4 |
| 15 | (DHQD)₂Pyr | CHCl₃ | 24 | 82 | − 45 |
| 16 | β-ICD | THF | 1 | 85 | 95 |
| 17 | β-ICD | EtOAc | 1 | 86 | 95 |
| 18 | β-ICD | CH₃CN | 1 | 81 | 95 |
| 19 | β-ICD | Toluene | 1 | 87 | 93 |
| 20 | β-ICD | CH₂Cl₂ | 1 | 82 | 95 |
| 21 | β-ICD | Ether | 1 | 83 | 90 |
| 22‡ | **β-ICD** | **EtOAc** | **24** | **96** | **96** |
| 23§ | β-ICD | EtOAc | 96 | 75 | 96 |

DABCO, 1,4-diazabicyclo[2.2.2]octane; DMAP, 4-Dimethylaminopyridine, (DHQ)₂PHAL, Hydroquinine 1,4-phthalazinediyl diether, and (DHQD)₂Pyr, Hydroquinidine-2,5-diphenyl-4,6-pyrimidinediyl diether. Reaction conditions: **1a** (0.15 mmol), the catalyst (0.015 mmol) in the solvent specified (3 ml) at room temperature.
*Yield of isolated product.
†Determined by HPLC analysis on a chiral stationary phase.
‡The loading of β-ICD was 1 mol%.
§The loading of β-ICD was 0.5 mol%.

linear, branched, aryl-bearing or ester-containing, and the corresponding alkyl-substituted α-allenic-γ-butyrolactones **2** were obtained in very high yields and with excellent enantiomeric excesses (Fig. 3, **2a–2h**). When the aryl-substituted cyclohexadienones were utilized, the reaction worked equally well, and the desymmerization products with excellent chemical yields and ee values were attainable (Fig. 3, **2i–2n**). In addition, dienone **1o** with an ethynyl group was found to be an equally suitable substrate. Moreover, we also employed dienone derived from 3,4,5-trimethylphenol, and bicyclic lactone **2p** bearing two contiguous quaternary stereogenic centres were obtained in moderate yield and 96% ee value after one recrystallization. It was

noteworthy, that only one single diastereomer was obtained in all cases. The absolute configurations of the allenoate RC products were determined based on X-ray crystal structural analysis of **2a** and **2n**.

**Dynamic kinetic resolution of racemic allenes.** The axially chiral allenes represent important structural motifs that are widely present in numerous bioactive molecules and natural products, and they are also versatile intermediates in synthetic organic chemistry[47–50]. In spite of the importance of chiral allenes, enantioselective synthesis of this class of compounds via

**Figure 3 | Scope of the intramolecular allenoate RC reaction catalyzed by β-ICD.** Reactions were performed with **1** (0.15 mmol) and β-ICD (0.0015 mmol) in EtOAc (3 ml) at room temperature. [a]The reaction was run at 0 °C with 5 mol% of β-ICD. [b]After once recrystallization. RT, room temperature.

organocatalytic methods[51–54] is less common. We were curious to find out whether it is feasible to access enantiomerically enriched allenes via a kinetic resolution, and we reasoned γ-substituted allenes may be suitable precursors for such a process. Toward this end, a racemic allene **6a** with a γ-substituted methyl group was prepared and subjected to β-ICD for further reaction. Indeed, a DKR process via an intramolecular allene RC reaction was observed. The DKR of **6a** at room temperature afforded chiral allene **7a** in 90% yield, and with poor diastereoselectivity (3:2) and excellent enantioselectivity (96% ee for the major diastereomer) (Table 2, entry 1). By lowering the reaction temperature to −5 °C, the diastereoselectivity of the reaction was increased to 5.5:1, and the enantioselectivity was further improved to 99% (entry 2). The scope of the observed DKR process was quite general. The γ-substituents of the starting racemic allenes can be varied, both linear and branched alkyl groups could be used, and good diastereoselectivities and excellent enantioselectivities were attainable (entries 3–7). Notably, the diastereomers were readily separable by flash column chromatography. Branched γ-substituents led to much improved diastereoselectivity (entries 4 and 7). However, allene **6g** with a very bulky *t*-butyl group was found to be unsuitable (entry 8). The R substituent at the cyclohexadienone ring can also be varied, and the corresponding allene was obtained in high enantiomeric excess (entry 9). The absolute configurations of chiral allenes **7** were determined based on the X-ray crystal structure of **7b**. Interestingly, when racemic cyclohexadienone **8** was utilized as a starting material, a parallel kinetic resolution process was observed, and the intramolecular RC product **9** with two contiguous quaternary centres was obtained with virtually perfect enantioselectivity (Fig. 4). To the best of our knowledge, the above DKR and parallel KR processes via intramolecular allene RC reaction are unprecedented, and may trigger more investigations for similar types of transformations.

To gain some insights into this interesting DKR process, we performed a few experiments to understand the dynamic behaviours of the reaction (Fig. 5). The RC-inactive substrate

**Table 2 | Dynamic kinetic resolution of racemic allenes 6.**

| Entry | 7 (R/R′) | Yield (%)* | dr† | ee (%)‡ |
|-------|----------|-----------|-----|---------|
| 1§ | **7a** (Me/Me) | 90 | 3:2 | 96 |
| 2 | **7a** (Me/Me) | 90 | 5.5:1 | 99 |
| 3 | **7b** (Me/Et) | 92 | 7:1 | 99.5 |
| 4 | **7c** (Me/i-Pr) | 93 | >19:1 | 98 |
| 5 | **7d** (Me/n-Bu) | 91 | 10:1 | 97 |
| 6 | **7e** (Me/CH₂CH₂Ph) | 90 | 16:1 | 98 |
| 7‖ | **7f** (Me/c-hexyl) | 85 | >19:1 | 98 |
| 8 | **7g** (Me/t-Bu) | – | – | – |
| 9 | **7h** (Ph/i-Pr) | 91 | 7.5:1 | 98 |

Reaction conditions: racemic **6** (0.15 mmol) and β-ICD (0.0075 mmol) in EtOAc (3 ml) at −5 °C for 24 h.
*Isolated yield.
†Determined by crude $^1$H NMR analysis.
‡The ee value of the major diastereomer, determined by HPLC analysis on a chiral stationary phase.
§The reaction was performed with β-ICD (1 mol%) at room temperature.
‖The reaction time was 48 h.

**Figure 4 | Parallel kinetic resolution.** Reactions were performed with **8** (36 mg, 0.18 mmol) and β-ICD (5 mol%, 3 mg) in EtOAc (3.6 ml) at −5 °C for 36 h.

**6g-1** in enantiomerically pure form via chiral HPLC separation was then treated with β-ICD at room temperature, and the resulting allenoate **6g-1′** was obtained with 69% ee, suggesting β-ICD-triggered racemization takes place before the RC reaction. The stereochemical integrity of the product bicyclic lactones was subsequently examined. When diastereomerically pure **7a** or **7c** was treated with β-ICD or DABCO respectively at room temperature, **7a′** or **7c′** with low diastereomeric ratio was obtained. When lactone **7c′** was exposed to β-ICD at −5 °C, no further erosion was observed, and the diastereomeric ratio remained virtually the same. The above base-induced epimerization of product lactones agrees well with our previous results in which only low dr ratio was attainable when the reaction was run at room temperature (Table 2, entry 1). A plausible mechanism is illustrated in Fig. 6. The reversible addition of β-ICD to allenoate **6** leads to the formation of zwitterionic intermediate **A**, and racemization takes place at this step. The RC reaction then follows, with the formation of intermediate **B**, and stereoselectivity of the reaction induced by chiral amine is established at this step. Subsequent proton transfer and elimination of β-ICD give rise to the formation of lactone product **7**. If the reaction is performed at room temperature, the stereochemical integrity of the product could not be maintained, and epimerization of the product occurs. However, performing reaction at −5 °C eliminates base-induced epimerization of optically enriched lactone products.

**Application in [4 + 2] annulation.** The allenes obtained via enantioselective intramolecular RC reactions are novel and interesting molecular architectures. The presence of the allenoate moiety certainly offers opportunities for further structural manipulations. As an illustration, when **2e** was reacted with α,β-unsaturated imine[55] in the presence of phosphine catalyst **12**, a novel [4 + 2] annulation reaction took place. The major diastereomer of tetrahydropyridine **13** was isolated in 70% yield and with 95% ee (Fig. 7).

## Discussion

In summary, we have developed the first enantioselective intramolecular RC reaction of allenoates, which enabled effective desymmetrization of cyclohexadienones, affording bicyclic α-allenic γ-butyrolactones in good chemical yields and excellent enantioselectivities. Furthermore, with the employment of dienone starting materials containing a γ-substituted allene moiety, an unprecedented DKR process was observed, yielding optically enriched chiral allenes. When a racemic cyclohexadienone was utilized, a novel parallel kinetic resolution process was discovered. Currently, other enantioselective desymmetrization processes of allene-containing substrates are being investigated in our laboratory and our findings will be reported in due course.

## Methods

**Materials.** For NMR spectra of compounds in this manuscript, see Supplementary Figs 1–55. For HPLC spectra of compounds in this manuscript, see Supplementary Figs 56–82. For the crystallographic data of compounds **2a**, **2n** and **7b**, see Supplementary Tables 1–3 and Supplementary Note 1. For the representative experimental procedures and analytic data of compounds synthesized, see Supplementary Methods.

**a**

**b**

**c**

**Figure 5 | Investigations of the DKR process.** (**a**) Optically pure RC-inactive substrate **6g-1** underwent racemization after treated with β-ICD. (**b**) Pure diastereomer **7a** epimerized upon treatment with β-ICD. (**c**) Epimerization of **7c** when treated by DABCO and β-ICD. RT, room temperature.

**Figure 6 | Proposed reaction mechanism.** A plausible mechanism illustrating how the desired products are formed and undesired racemization processes.

**Figure 7 | The [4 + 2] annulation of the allenoate RC product 2e with imine 11.** Reaction was performed with **2e** (0.1 mmol, 23 mg), imine **11** (0.15 mmol, 51 g) and **12** (0.015 mmol, 7.2 mg) in toluene (1 ml) at RT. RT, room temperature.

**Preparation of 2.** To a flame dried round bottle flask with a magnetic stirring bar at room temperature under $N_2$ were added allenoate **1** (0.15 mmol) and ethyl acetate (3 ml), followed by the addition of β-ICD (1 mol%, 0.5 mg). The resulting mixture was stirred at room temperature for 24 h. The solvent was removed under reduced pressure and the residue was purified by column chromatography on silica gel to afford annulation adduct **2**.

**Data availability.** The X-ray crystallographic coordinates for structures that support the findings of this study have been deposited at the Cambridge Crystallographic Data Centre (CCDC) with the accession code CCDC 1444859 (**2a**), 1444860 (**2n**) and 1444861 (**7b**) (www.ccdc.cam.ac.uk/data_request/cif). The authors declare that all other data supporting the findings of this study are available within the article and Supplementary Information files, and also are available from the corresponding author upon reasonable request.

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

## Acknowledgements

We thank Singapore A*STAR SERC Public Sector Research Funding (PSF; R-143-000-618-305), the National University of Singapore (R-143-000-599-112), and GSK–EDB (R-143-000-491-592) for generous financial support.

## Author contributions

W.Y. planned and conducted most of the experiments; X.D. and S.W. prepared substrates for the reaction scope evaluation; J.W. analysed the two-dimension NMR spectrum; J.J.V. analysed the X-ray single crystal data; Y.L. conceptualized and directed the project and Y.L. and W.Y. co-wrote the manuscript. All authors contributed to the discussion.

## Additional information

**Competing financial interests:** The authors declare no competing financial interests.

**How to cite this article**: Yao, W. *et al.* Enantioselective desymmetrization of cyclohexadienones via an intramolecular Rauhut–Currier reaction of allenoates. *Nat. Commun.* **7,** 13024 doi: 10.1038/ncomms13024 (2016).

