## [Peer review file · Nature Communications]

Reviewers' comments:

Reviewer #1 (Remarks to the Author):

This paper describes an enantioselective intramolecular Rauhut-Currier reaction of prochiral cyclohexadienone derivatives bearing an allenolate moiety. Utilizing β -ICD as an organocatalyst, high yield of products are obtained in good to excellent diastereoselectivities and enantioselectivities. It is noteworthy that the reaction proceeds very mild conditions in environmentally benign solvent and the scope of the reaction is reasonably broad. The authors also achieved the first dynamic kinetic resolution of the enantioselective intramolecular R-C reaction using racemic γ -substituted allenolate derivatives. The reviewer believes this manuscript is worth publishing in Nature Communications almost as is. Some small typographical errors (e.g. in the legend of Figure 3: super script "c" must be "b" and the length of the hyphen in entry 9 of Table 1) should be corrected.

Reviewer #2 (Remarks to the Author):

The Rauhut-Currier (RC) reaction is one of the fundamental reaction for carbon-carbon bond formation, it's also an effective atom-economic method to construct densely functionalized molecules. However, only two examples of the intermolecular Rauhut-Currier reactions of allenolates and active alkene were reported. This paper(NCOMMS-16-02956) describes an excellent achievement on the intramolecular Rauhut-Currier (RC) reaction of the intramolecular reaction of allenolates. In the presence of 1 mol% β -ICD, the RC reaction proceeded smoothly, and bicyclic lactones were obtained in high yields and with excellent enantiomeric excesses. With the employment of γ -substituted allenolates as racemic precursors, a novel dynamic kinetic resolution of allenes via RC reaction was observed, a simple application of the corresponding product is also tested. This method provides a new strategy for the synthesis of bicyclic α -allenic γ -butyrolactones, it's novel not only for the RC reaction but also for the allenolate products, for this reason, I believe that the current work merits publication in the Nat. Comm. after the following points have been addressed:

1. About the introduction, a brief introduction of the Rauhut-Currier (RC) reaction should be included.

2. About the reference,

1). for reviews on the RC reaction: a) J. L. Methot, W. R. Roush, *Adv. Synth. Catal.* 2004, 346, 1035-1050; b) P. Xie, Y. Huang, *Eur. J. Org. Chem.* 2013, 6213-6226 should be cited.

2). The recent report on highly enantioselective intermolecular RC reaction (*Angew. Chem. Int. Ed.* 2015, 54, 1621-1624, *Angew. Chem. Int. Ed.* 2015, 54 (49), 14853-14857) should be cited.

3). In the reference 43, 48, 50, the full authors should be added.

3. About the data of the X-Ray crystallographic data of the product not good enough for publication, all X-Ray crystallographic data of the three products have a serious problem, it need resolve or explain.

Reviewer #3 (Remarks to the Author):

A - Summary: The manuscript by Lu et al. describes the use of a cinchona-derived tertiary amine (β -ICD) as catalyst for the stereoselective cyclization of allenolates via Rauhut-Currier reaction. The transformation yields bicyclic γ -lactones and the protocol appears quite generally applicable with a good substrate scope and high optical purities. In addition, starting from substituted racemic allenes a dynamic kinetic asymmetric transformation is observed, giving rise to axially chiral allenic products.

B - Originality and interest: The presented work is novel, however, perhaps a bit too specialized for Nature Communications.

C - Data & methodology: All presented data are supported by solid supporting information. The presentation, however, could certainly be improved with regard to phrasing and experiments (see F).

D - Appropriate use of statistics and treatment of uncertainties: fine.

E - Conclusions: robustness, validity, reliability: fine.

F - Suggested improvements: The manuscript presents convincingly the development of a novel organocatalytic method that is certainly of interest for organic chemists. However, in my opinion the presented data appear somewhat incomplete. As a conclusion of the optimization Table 1, beta-ICD was identified as effective catalyst. 96% ee is without doubt a good result, but how about other tertiary amine organocatalysts? It is probably not necessary to artificially expand Table 1, though, I would expect that the more obvious structural analogues (other cinchona alkaloids, (DHQ)2PHAL and variations thereof) were tested, and should be included. Moreover, while the outcome of the so-called dynamic resolution (in my eyes, this is more of a dynamic asymmetric transformation since resolutions do not alter the substrate structure this dramatically) is very promising, there is no information on the actual dynamic behaviour. Usually DKR studies involve elucidation of the racemization kinetics in order to understand and optimize the overall process. In this case, it is not even clear whether racemization is taking place prior to the Rauhut-Currier reaction, or if the increased diastereoselectivity occurs as a result of a subsequent epimerization of the chiral axis. Experiments on the racemization of Rauhut-Currier-inactive allenates by beta-ICD and on the stereochemical integrity vs epimerization of the product lactones would be required in addition to these otherwise appealing results.

G - References: too long.

H - Clarity and context: could be improved (see F)

Reviewer #4 (Remarks to the Author):

In this manuscript, the desymmetrization of cyclohexadienones bearing an allene moiety via enantioselective intramolecular Rauhut-Currier reaction of allenates has been successfully developed. This is an interesting paper and is important to the field. The authors do have some good chemistry; however X-ray crystal structure results were roughly prepared:

There are some alerts A/B can be found in CHECKCIF without any explanations. The H atoms for 2a ("CCDC 1444859" in cif; "e488" in SI) and 2n ("CCDC 1444860" in cif; "f130" in SI), in fact, were not refined appropriately. The Flack parameter for 7c ("CCDC 1444861" in cif; "f222" in SI) is up to 0.10(5), which may not give an UNEQUIVOCAL result for the absolute configuration.

In fact, the author should consider re-refining their structure with Shelxl2014 with version 2014/7 and including the data (.hkl) in their .cif. In this case, any readers who interests in the crystal structure will be able to download their data sets and refinement result file from the cif.

The relevant reference should be updated

Point to Point Response to Reviewers' Comments

Please take note that all the descriptive, positive comments of the reviewers are omitted, and only the reviewers' comments expressing their concerns/suggestions are listed below, which are followed by our responses.

Reviewer 1' comments and our responses

Comments: The reviewer believes this manuscript is worth publishing in Nature Communications almost as is. Some small typographical errors (e.g. in the legend of Figure 3: super script "c" must be "b" and the length of the hyphen in entry 9 of Table 1) should be corrected.

Response: The above typographical errors pointed out by reviewer 1 have been corrected, i.e. correct the super script "c" to "b" in Figure 3; change "β–ICD" to "β-ICD" in Table 1.

Reviewer 2' comments and our responses

Comments: I believe that the current work merits publication in the Nat. Comm. after the following points have been addressed: 1. About the introduction, a brief introduction of the Rauhut-Currier (RC) reaction should be included.

Response: A short introduction to Rauhut–Currier has now been added (Page 2, revised manuscript).

Comments: 2. About the reference, 1). for reviews on the RC reaction: a) J. L. Methot, W. R. Roush, Adv. Synth. Catal. 2004, 346, 1035-1050; b) P. Xie, Y. Huang, Eur. J. Org. Chem. 2013, 6213-6226 should be cited. 2). The recent report on highly enantioselective intermolecular RC reaction (Angew. Chem. Int. Ed. 2015, 54, 1621-1624, Angew. Chem. Int. Ed. 2015, 54 (49), 14853-14857) should be cited. 3). In the reference 43, 48, 50, the full authors should be added.

Response: The two review articles on the RC reaction, and two recent reports have been added (refs 5 & 7, and refs 8 & 9 in the revised manuscript). Since the numbers of the authors for all those references pointed out by reviewer 3 (refs. 43, 48, 50 in the original submission) are ≥ 6 , we thus list only the first authors followed by et al. following the submission guidelines (designated as refs. 39, 44, and 46 in the revised manuscript).

Comments: 3. About the data of the X-Ray crystallographic data of the product not good enough for publication, all X-Ray crystallographic data of the three products have a serious problem, it need resolve or explain.

Response: We have refined all the structures (**2a**, **2n**, and **7b** in the revised manuscript), and all the CHECKCIFs are free from alerts A and B (please also see our responses to reviewer 4 in the subsequent part of this document).

Reviewer 3' comments and our responses

Comments: F - Suggested improvements: The manuscript presents convincingly the development of a novel organocatalytic method that is certainly of interest for organic chemists. However, in my opinion the presented data appear somewhat incomplete.

- As a conclusion of the optimization Table 1, beta-ICD was identified as effective catalyst. 96% ee is without doubt a good result, but how about other tertiary amine organocatalysts? It is probably not necessary to artificially expand Table 1, though, I would expect that the more obvious structural analogues (other cinchona alkaloids, (DHQ)2PHAL and variations thereof) were tested, and should be included.

Response: Following the above suggestion, we examined catalytic effects of some common cinchona alkaloids, including (DHQ)2PHAL, and the corresponding results were summarized in the revised Table 1 (entries 10–15, Table 1, the revised manuscript).

Comments:

- Moreover, while the outcome of the so-called dynamic resolution (in my eyes, this is more of a dynamic asymmetric transformation since resolutions do not alter the substrate structure this dramatically) is very promising, there is no information on the actual dynamic behaviour. Usually DKR studies involve elucidation of the racemization kinetics in order to understand and optimize the overall process. In this case, it is not even clear whether racemization is taking place prior to the Rauhut-Currier reaction, or if the increased diastereoselectivity occurs as a result of a subsequent epimerization of the chiral axis. Experiments on the racemization of Rauhut-Currier-inactive allenates by beta-ICD and on the stereochemical integrity vs epimerization of the product lactones would be required in addition to these otherwise appealing results.

Response: Following the above valuable suggestions made by reviewer 3, we performed the following experiments:

- We studied racemization of enantiomerically pure RC-inactive allenate **6g-1** (obtained via chiral HPLC separation) by β -ICD (summarized in Figure 5 in the revised manuscript);
- We treated diastereomerically pure product lactones **7a** and **7c** with β -ICD and DABCO, respectively, to establish stereochemical integrity vs epimerization of the product lactones (summarized in Figure 5 in the revised manuscript);
- The above experiments suggested racemization takes place prior to the RC reaction (see discussion on pages 5 & 6 in the revised manuscript);
- We also proposed reaction mechanism based on our investigations (see page 6 and Figure 6 in the revised manuscript).

Comments: G- References: too long.

Response: the number of references in the current submission is 55, no more than 70, abiding to the submission guidelines.

Reviewer 4' comments and our responses

Comments: The authors do have some good chemistry; however X-ray crystal structure results were roughly prepared: There are some alerts A/B can be found in CHECKCIF without any explanations. The H atoms for 2a ("CCDC 1444859" in cif; "e488" in SI) and 2n ("CCDC 1444860" in cif; "f130" in SI), in fact, were not refined appropriately. The Flack parameter for 7c ("CCDC 1444861" in cif; "f222" in SI) is up to 0.10(5), which may not give an UNEQUIVOCAL result for the absolute configuration. In fact, the author should consider re-refining their structure with Shelxl2014 with version 2014/7 and including the data (.hkl) in their .cif. In this case, any readers who interests in the crystal structure will be able to download their data sets and refinement result file from the cif.

Response: Following the above comments, we have refined these structures (**2a**, **2n** and **7b** instead of **7c**) again using Shelx 2014 and the new CIFs and CHECKCIF files are attached. After the refinements, **all the CHECKCIFs are free from alerts A and B**. Since the X-ray crystal data for **7c** (in the original submission) (CCDC 1444861) was not of high quality, we performed single X-ray crystallographic studies of another crystalline compound (**7b**, CCDC 1484783).

Comments: The relevant reference should be updated.

Response: We have updated all the references to the best we can (see refs 5, 7, 8 and 9 in the revised manuscript).

REVIEWERS' COMMENTS:

Reviewer #3 (Remarks to the Author):

In the revised manuscript, the authors have addressed all comments by the reviewers in a convincing manner. I particularly appreciate the effort to include the suggested mechanistic/stereochemical studies that in my opinion really added further value to the article. Just a quick addition to figure 6: as correctly pointed out, reversible betain formation from the starting material explains the required racemization to allow for the observed dynamics, however, it would be nice to also address the epimerization issue in the proposed mechanism simply by a small "(undesired epimerization at room temperature)" under the last reversible arrow that leads to the RC product.

Other than that, I fully support publication of this work in its current form in Nature Communications.

Reviewer #4 (Remarks to the Author):

In this revised manuscript, the authors have re-refined their structures; however, I note that:

1) The authors did not describe how they perform their X-ray crystallography experiment and refine their structure (including reference) in manuscript or SI.

2) In data 2a, the Ueq for some H atoms have been set as 21.00000 without any expatiation:

AFIX 0

H10C 2 0.258895 0.439497 -0.040383 11.00000 21.00000

H10B 2 -0.624147 0.773375 0.981043 11.00000 21.00000

H10A 2 -0.596843 0.964908 0.879993 11.00000 21.00000
H10D 2 0.118235 0.446847 0.130996 11.00000 21.00000

Such approach is very strange for SHELXL refinement. Similar things can be found in data_2n:
H20B 2 0.399552 0.208002 0.513789 11.00000 21.00000
H20A 2 0.323545 0.329797 0.564404 11.00000 21.00000

To learn how to refine their structure correctly, the authors may consider carefully reading through Chapter 3 Hydrogen Atoms in Peter Muller's Crystal Structure Refinement, A Crystallographer's Guide to SHELXL.

Point to Point Response to Reviewers' Comments

Please take note that all the descriptive, positive comments of the reviewers are omitted, and only the reviewers' comments expressing their concerns/suggestions are listed below, which are followed by our responses.

Reviewer 3' comments and our responses

Comments: Just a quick addition to figure 6: as correctly pointed out, reversible betain formation from the starting material explains the required racemization to allow for the observed dynamics, however, it would be nice to also address the epimerization issue in the proposed mechanism simply by a small "(undesired epimerization at room temperature)" under the last reversible arrow that leads to the RC product.

Response: We have revised Figure 6 accordingly, a small arrow is used to indicate the undesired epimerization at room temperature (please see revised Figure 6).

Reviewer 4' comments and our responses

Comments: In this revised manuscript, the authors have re-refined their structures; however, I note that:

1) The authors did not describe how they perform their X-ray crystallography experiment and refine their structure (including reference) in manuscript or SI.

Response: As reviewer requested, the following paragraph describing X-ray crystallography experiments and refinements of the structures were added in the SI (see Supplementary Note in the SI, page 84). In addition, the relevant references were added (see Supplementary References in the SI).

Paragraph appearing on page 84 of the SI describing X-ray experiments and refinements: "Crystal data of all these crystals CCDC No. 1444859 (2a), 144860 (2n) and 1444861 (7b) were collected on a Bruker AXS D8 Venture equipped with a Photon 100 CMOS active pixel sensor detector using graphite-monochromated Cu-K α radiation ($\lambda = 1.54178 \text{ \AA}$) using a sealed tube. Absorption

corrections were made with the program SADABS,¹ and the crystallographic package SHELXL 2, 3 was used for all calculations”.

References listed in the Supplementary References are the follows:

1. Sheldrick, G. M., University of Göttingen, Germany, 1996.
2. Sheldrick, G. M. A short history of SHELX. *Acta Crystallogr. Sect. A.* **64**, 112-122 (2008).
3. Müller, P., Herbst-Irmer, R., Spek, A. L., Schneider, T. & Sawaya, M. *Crystal Structure Refinement: A Crystallographer's Guide to SHELXL*. Chapter 3, Oxford, UK, Oxford University Press/International Union of Crystallography, 2006.

Comments: 2) In data 2a, the Ueq for some H atoms have been set as 21.00000 without any expatiation:

AFIX 0

H10C 2 0.258895 0.439497 -0.040383 11.00000 21.00000

H10B 2 -0.624147 0.773375 0.981043 11.00000 21.00000

H10A 2 -0.596843 0.964908 0.879993 11.00000 21.00000

H10D 2 0.118235 0.446847 0.130996 11.00000 21.00000

Such approach is very strange for SHELXL refinement. Similar things can be found in data_2n:

H20B 2 0.399552 0.208002 0.513789 11.00000 21.00000

H20A 2 0.323545 0.329797 0.564404 11.00000 21.00000

To learn how to refine their structure correctly, the authors may consider carefully reading through Chapter 3 Hydrogen Atoms in Peter Muller's *Crystal Structure Refinement, A Crystallographer's Guide to SHELXL*.

Response: We have carefully read through the Chapter 3 Hydrogen Atoms in Peter Muller's *Crystal Structure Refinement, A Crystallographer's Guide to SHELXL*, as advised by reviewer 4, before carrying out the refinements again. As per the book, we have refined the riding model for the Ueq by replacing 21.000 by -1.2000. We have also rearranged the H atoms and put them next to the respective C atoms. The new CIFs have been uploaded. We sincerely thank the crystallographer for his advice.